# Alimentary Treatment with Trehalose in a Pharmacological Model of Alzheimer’s Disease in Mice: Effects of Different Dosages and Treatment Regimens

**DOI:** 10.3390/pharmaceutics16060813

**Published:** 2024-06-16

**Authors:** Alexander B. Pupyshev, Anna A. Akopyan, Michael V. Tenditnik, Marina V. Ovsyukova, Nina I. Dubrovina, Victor M. Belichenko, Tatiana A. Korolenko, Svetlana A. Zozulya, Tatiana P. Klyushnik, Maria A. Tikhonova

**Affiliations:** 1Laboratory of the Neurobiological Mechanisms of Neurodegenerative Processes, Department of Experimental Neuroscience, Scientific Research Institute of Neurosciences and Medicine (SRINM), 630017 Novosibirsk, Russia; 2Mental Health Research Center, 115522 Moscow, Russia; s.ermakova@mail.ru (S.A.Z.);

**Keywords:** Alzheimer’s disease, amyloid-β 25–35, hippocampus, autophagy, trehalose, LC3-II, microglia, passive avoidance

## Abstract

In the treatment of experimental neurodegeneration with disaccharide trehalose, various regimens are used, predominantly a 2% solution, drunk for several weeks. We studied the effects of different regimens of dietary trehalose treatment in an amyloid-β (Aβ) 25–35-induced murine model of Alzheimer’s disease (AD). Aβ-treated mice received 2% trehalose solution daily, 4% trehalose solution daily (continuous mode) or every other day (intermittent mode), to drink for two weeks. We revealed the dose-dependent effects on autophagy activation in the frontal cortex and hippocampus, and the restoration of behavioral disturbances. A continuous intake of 4% trehalose solution caused the greatest activation of autophagy and the complete recovery of step-through latency in the passive avoidance test that corresponds to associative long-term memory and learning. This regimen also produced an anxiolytic effect in the open field. The effects of all the regimens studied were similar in Aβ load, neuroinflammatory response, and neuronal density in the frontal cortex and hippocampus. Trehalose successfully restored these parameters to the levels of the control group. Thus, high doses of trehalose had increased efficacy towards cognitive impairment in a model of early AD-like pathology. These findings could be taken into account for translational studies and the development of clinical approaches for AD therapy using trehalose.

## 1. Introduction

The multifactorial nature of Alzheimer’s disease (AD), involving different molecular and cellular mechanisms in the disease progression, justifies the search for multifunctional therapeutic agents or combinations of drug effects [1,2]. A promising candidate substance for this is the disaccharide trehalose, which has shown efficacy in inhibiting experimental neurodegeneration in several models of AD [3,4]. Several mechanisms of therapeutic action have been identified [5,6,7]. The drug inhibits the accumulation of the AD marker amyloid-β (Aβ); exhibits chaperone-like properties that prevent the formation of protein aggregates; weakens oxidative stress; promotes endogenous antioxidant protection; stimulates autophagy, lysosomal autophagy flux, and lysosomal biogenesis; and enhances the removal of cytotoxic protein material. The activation of autophagy seems to be the main therapeutic activity [3,5]. The activation of autophagosomal segregation and the removal of protein material with cytotoxic properties, such as hyperphosphorylated tau or Aβ aggregates [8], as well as damaged mitochondria and other abnormal intracellular entities [9,10], have been revealed in the trehalose-treated AD models. Importantly, the experimental suppression of autophagy largely inhibits the neuroprotective effect of trehalose [11,12]. The therapeutic effect of trehalose, ultimately assessed by the restoration of behavioral responses, has also been reproduced in transgenic models of chronic neurodegeneration, during long-term treatment of the neurodegenerative diseases with the drug [13,14].

Regarding the high translational potential of trehalose, certain issues related to optimal conditions for its clinical application attract particular attention. The safety of taking high doses of trehalose [15,16], the efficiency of the oral and parenteral administration of the drug [17], the dose dependence of the therapeutic effect of the drug [18], the effects of continuous treatment [13,17] or intermittent regimen [19], as well as the age-dependence, [13] have been researched. The studied concentrations of trehalose solution in the diet range from 1% to 5%, with different amplitudes of the achieved autophagic response [18,20,21,22,23]. A 2% concentration of trehalose in the drinking water is considered standard to induce neuroprotective effects in laboratory rodents [18,24]. Some studies disclosed the advantages of high doses of trehalose due to its enhanced penetration into the blood [21,22]. On the other hand, the persistent induction of autophagy could be harmful, leading to cell self-destruction and loss [25,26]. Hence, the potential of intermittent treatment with high doses of trehalose in experimental models of AD are of interest [3,19]. According to a recent a systematic review [4], translational studies of trehalose will require an evaluation of the optimal “time window” and dosage to confer neuroprotective benefits.

The aim of this work was to evaluate the dose dependence and regimens of trehalose treatment (continuous vs. intermittent) on the molecular and cellular markers of neuroprotection and the recovery of cognitive behavioral responses in a pharmacological model of AD induced by i.c.v. administration of the Aβ25–35 fragment.

## 2. Materials and Methods

### 2.1. Experimental Animals and Procedures Involving Animals

Male C57BL/6 mice (2.5 months old, 25–30 g) from the Federal State Budgetary Scientific Institution *Scientific Research Institute of Neurosciences and Medicine* (SRINM; Novosibirsk, Russia) were used. The animals were maintained on a standard laboratory diet under standard conditions (the light–dark cycle: 14 h light and 10 h dark, temperature: 20–22 °C, and relative humidity: 50–60%). Every effort was made to minimize the number of animals used and their suffering. All protocols follow the recommendations outlined in the NIH Guidelines for the Care and Use of Laboratory Animals (National Institutes of Health Publication, 8th edition, revised 2011) and were approved by the Ethics Committee of the Scientific Research Institute of Neurosciences and Medicine (ID 2, date of approval: 21 February 2019).

The experiments were carried out on a murine pharmacological model of AD induced by the central administration of the human amyloid beta fragment oligomers Aβ25–35 (Amyloid Beta-Protein Fragment 25–35 (Sigma-Aldrich, Darmstadt, Germany)). Amyloid preparation and the injections into cerebral ventricles were performed according to previously published protocols [27]. A total of 5 μL of the drug or vehicle were administered into each lateral ventricle using Neurostar’s Stereo Drive motorized stereotaxis system (Neurostar, Tübingen, Germany). A total of 10 μL (9.43 nmole) of the solution was administered to each mouse. The following coordinates adapted from the mouse brain atlas [28] were used: AP = −0.46 mm; ML = ±0.9 mm; and DV = 2.5 mm from the bregma, midline, and skull surface, respectively.

The treatment with trehalose started 2 days after the stereotaxic surgery. The practice of treating mice with trehalose, using its 1–5% solutions as a drink, is widespread. In our experiments, the amount of liquid drunk was determined. Mice consumed 5.63 ± 0.18 mL of water or 5.14 ± 0.30 mL of trehalose solution daily per mouse. The mice were subdivided into five groups (6 to 9 animals each): (1) control (bilateral injections of sterile water into the lateral ventricles of the brain [hereafter: i.c.v. injection]); (2) Aβ (bilateral i.c.v. injections of Aβ25–35, AD model); (3) “Aβ + 2% T” (AD model + a 2% solution of trehalose in drinking water daily for 2 weeks); (4) “Aβ + 4% T” (AD model + a 4% solution of trehalose in drinking water daily for 2 weeks); (5) “Aβ + 4% T/2” (AD model + a 4% solution of trehalose in drinking water every other day for 2 weeks). One day after the final drug administration, behavioral testing was performed, after which biological samples were collected from the randomly selected six mice of each group.

### 2.2. Behavioral Testing

Each mouse was handled for 5 min/day during two consecutive days before the behavioral testing. The animals were kept in groups of 3–6 mice per cage between tests. Prior to testing, a mouse was placed individually in a clean cage (25 cm × 40 cm × 20 cm) and transported to a dimly lit observation room (28 lux), with sound isolation reinforced by a masking white noise of 70 dB. A test was started 10 min after establishing the habituation of a mouse in line the testing conditions. Animal behavior was recorded using two digital video cameras Panasonic WV-CL930 (Panasonic System Networks Suzhou Co. Ltd., Suzhou, China) from the side and top. The video data were processed in original EthoVision XT 11.5 software (Noldus IT, Wageningen, The Netherlands). The test equipment was cleaned with 20% ethanol and thoroughly dried before each test trial.

*The open field test:* This test was carried out to assess locomotor and exploratory activity and anxiety. The following parameters were measured: general locomotor activity by the total distance traveled (cm); anxiety by the time spent in the center of the arena; vertical locomotor activity and exploratory behavior by the number of rearing bouts. The test was performed in an open-top transparent Plexiglas installation with a square arena (40 × 40 × 37.5 cm) under bright illumination (1000 lux) from above. Mice were placed in the arena near the wall and their behavior was recorded for 10 min.

*Passive avoidance test:* Training on the passive avoidance response was performed by a standard single-session method in an experimental chamber with dark and light compartments and an automated Gemini Avoidance System apparatus (San Diego Instruments, San Diego, CA, USA), as described in detail earlier [12]. The Gemini software Version 1.0.0 (San Diego Instruments, San Diego, CA, USA) automatically recorded the latency when entering the dark compartment, and the data from the test served as a measure of the acquisition of the conditioned passive avoidance response.

### 2.3. Neuromorphological Analysis

Mice were culled by CO_2_ asphyxiation and transcardially perfused with phosphate-buffered saline (PBS), followed by 4% paraformaldehyde in PBS; then, their brains were rapidly removed and postfixed in PBS containing 30% of sucrose at 4 °C. After dehydration, brain samples were embedded in Tissue-Tek O.C.T. compound (Sakura Finetek, Torrance, CA, USA) and frozen at −70 °C, until sectioning into 30-μm-thick slices on a cryostat HistoSafe MicroCut—SADV (Citotest Scientific Co., Ltd., Nanjing, China). Coronal slices along the frontal cortex [AP = 2.93–2.57 mm] or hippocampus [AP = −1.91–(−2.45) mm] of each mouse brain were carried out. The immunohistochemical analysis (IHC) and Nissl staining were performed according to protocols described in detail previously [12,27].

### 2.4. Assay for Leukocyte Elastase (LE) Activity

Leukocyte elastase (LE) activity was measured by the rate of cleavage of the chromogenic substrate Boc-l-alanine 4-nitrophenyl ester (BOC-Ala-ONp) (ICN Biomedical Inc., Costa Mesa, CA, USA), as described in detail previously [29]. The incubation mixture included 0.480 mL PBS (pH 6.5), 0.02 mL 0.40% BOC-Ala-ONp substrate solution (3.1 mg in 1 mL acetonitrile), and 0.005 mL of the serum to be tested. Using the SWIFT 1000 Reaction Kinetics software (version 2.03, Biochrom Ltd., Cambridge, UK), optical density (OD) was measured at 347 nm for 3 min.

### 2.5. Assay for α1-Protease Inhibitor (α1-PI) Activity

α1-protease inhibitor (α1-PI) is the main LE inhibitor in the blood. Since α1-PI is a nonspecific inhibitor of serine proteases, its functional activity in the serum was measured by an enzymatic method based on trypsin inhibition using *N*-α-benzoyl-l-arginine ethyl ether hydrochloride (BAEE) as a substrate (ICN Biomedical Inc., Costa Mesa, CA, USA) [29]. The experimental sample consisted of 1.9 mL Tris-HCl buffer (pH 8.0); 0.1 mL serum diluted with saline (1:50); and 0.01 mL 0.01% trypsin solution. After preincubation for 3 min at 25 °C, 1 mL of 1.5 mM BAEE solution was added to the sample and then the increase in OD at 253 nm was measured for 3 min using the SWIFT 1000 Reaction Kinetics computer program (version 2.03, Biochrom Ltd., Cambridge, UK).

### 2.6. Statistics

STATISTICA 10.0 software (StatSoft, Tulsa, OK, USA) was used to perform all the statistical analyses. The normality of the data distribution was determined by the Shapiro–Wilk *W* test. The statistical evaluation of data was performed by one-way analysis of variance (ANOVA) or repeated measures ANOVA (results of the passive avoidance test) followed by Fisher’s LSD post hoc test. The level of significance was defined as *p* < 0.05 in all experiments reported here.

## 3. Results

### 3.1. Autophagy Activity

Autophagy activity was evaluated by an IHC analysis of autophagy marker LC 3-II in the brain areas. We evaluated the immunofluorescence of the LC3B antigen to examine the activity of autophagy in the frontal cortex and hippocampus. The results are summarized in Figure 1. The experimental groups significantly varied in LC3-II expression in all brain structures examined: frontal cortex [*F*(4, 10) = 13.78, *p* < 0.001], CA1 hippocampal area [*F*(4, 10) = 6.73, *p* < 0.01], CA3 hippocampal area [*F*(4, 10) = 9.21, *p* < 0.01], dentate gyrus (DG) [*F*(4, 10) = 14.51, *p* < 0.001]. Although there was a tendency to increase in autophagy activity in Aβ group compared to vehicle-treated i.c.v. controls in all the brain areas examined, a significant enhancement of the LC3-II expression was observed in the DG of the hippocampus (*p* < 0.05). An overall trend in increased LC3-II levels compared to the Aβ group was found in all the trehalose-treated groups. Nevertheless, only mice in the group treated with the highest dosage of trehalose (“Aβ + 4% T”, 4% trehalose solution daily) had significantly higher levels of LC3-II immunofluorescence than in the Aβ group in all the brain areas studied. Moreover, mice of the two other trehalose-treated groups were characterized by the reduced LC3-II expression compared to the “Aβ + 4% T” group. It is worth noting that the effect of the intermittent application of trehalose (“Aβ + 4% T/2”, 4% trehalose solution every other day) was markedly less than that of continuous treatment. Thus, a dose-dependent increase in the expression of autophagy marker LC3-II with trehalose treatment was revealed, with maximal efficiency for the continuous application of the 4% solution.

### 3.2. Aβ Accumulation

Aβ accumulation was evaluated in the frontal cortex and hippocampus. The groups differed significantly in Aβ levels in the frontal cortex [*F*(4, 10) = 37.42, *p* < 0.001] and hippocampal CA1 [*F*(4, 11) = 7.19, *p* < 0.01] and CA3 [*F*(4, 10) = 4.76, *p* < 0.05] areas. Aβ levels were significantly higher in the Aβ-treated group than in the controls in the frontal cortex (*p* < 0.001), hippocampal CA1 (*p* < 0.001), and CA3 (*p* < 0.01) areas (Figure 2). Aβ load was significantly attenuated by trehalose treatment in all modes of trehalose administration in the frontal cortex, hippocampal CA1 area, and hippocampal CA3 area. The changes in Aβ levels among the trehalose-treated groups were insignificant.

### 3.3. Neuronal Density (Nissl Staining)

The neuronal density, measured as the area occupied by Nissl-stained cells, was studied in the frontal cortex and hippocampus (Figure 3). The groups differed significantly in this parameter in the frontal cortex [*F*(4, 17) = 4.38, *p* < 0.01] and in the hippocampal CA1 area [*F*(4, 17) = 2.97, *p* < 0.05]. Neuronal density was significantly decreased in the frontal cortex (*p* < 0.001) and hippocampal CA1 area (*p* < 0.01) in the mice of the AD models (Aβ group) compared to the control group, while the trehalose-treated groups did not differ in this parameter from controls. Thus, trehalose prevented Aβ-linked neuronal loss in the frontal cortex and the CA1 area of the hippocampus. No significant differences were found between the mice given trehalose in different regimens.

### 3.4. Inflammatory Response

Brain neuroinflammatory response was evaluated by the microglia activation in the frontal cortex and hippocampus. Similar to Aβ accumulation, the expression of the microglial marker IBA1 varied significantly between the groups in the frontal cortex [*F*(4, 14) = 5.22, *p* < 0.01] and hippocampal CA1 [*F*(4, 14) = 3.23, *p* < 0.05], CA3 [*F*(4, 14) = 4.17, *p* < 0.05], or DG [*F*(4, 15) = 3.37, *p* < 0.05] areas. The AD model (Aβ group) was characterized by substantially augmented levels of IBA1 expression in all the brain areas studied, while trehalose treatment in all modes reduced the parameter down to the values of the control group. At the same time, microglial response did not differ significantly among the trehalose-treated groups. Thus, an inhibitory effect of trehalose on neuroinflammation in the brain was found.

It was of interest to further estimate the relationship between the therapeutic effect of trehalose and the suppression of inflammation using indices of systemic inflammatory response and immune status [30]. Some peripheral immune indices have been studied earlier, regarding the course and treatment of AD in humans [31]. The functional activity of blood neutrophils was analyzed by the activity of LE (Figure 4A). This parameter was not affected significantly by the i.c.v. administration of Aβ25–35, and neither was an acute phase protein α1-PI (Figure 4B). No significant effect of trehalose treatment on α1-PI levels was observed [*F*(4, 19) = 2.38, *p* > 0.05]. At the same time, the effects of trehalose on the activity of LE were inconsistent [*F*(4, 19) = 12.67, *p* < 0.001]. Consumption of the 2% trehalose solution enhanced the parameter, while the intermittent application of 4% trehalose solution every other day markedly reduced the activity of the enzyme.

### 3.5. Behavioral Effects

The efficacy of trehalose in restoring cognitive function was studied using the passive avoidance test. There was a significant effect as a result of the group factor [treatment; *F*(4, 25) = 14.94, *p* < 0.001], the learning factor [repeated measures; *F*(1, 25) = 345.54, *p* < 0.001], and the interaction between these factors [*F*(4, 25) = 15.64, *p* < 0.001] on step-through latency (Figure 5). The latency in entering a dark compartment during training (before the foot shock) did not differ significantly between experimental groups. As evidence of learning on the testing day, 24 h after receiving the foot shock, mice of the control group showed markedly longer step-through latency values than those on the training day (*p* < 0.001). In contrast, step-through latency on the testing day was sharply lower in the Aβ-treated group (AD model) than in the controls (*p* < 0.001), thereby indicating memory impairment. Trehalose successfully reversed these behavioral alterations. All three modes of trehalose treatment used produced a recovery of the learning and memory deficits, as evidenced by a significantly longer step-through latency compared to the Aβ-treated group (*p* < 0.001). A dose-dependent effect of trehalose was revealed, with maximal values in the group given 4% trehalose solution daily. Step-through latency on the test day was significantly higher in mice treated with 4% trehalose solution daily than in the other two trehalose-treated groups (*p* < 0.001). Moreover, the values of step-through latency were only restored up to the values of the controls in the “Aβ + 4% T” group. The effect of the intermittent regimen of 4% trehalose solution was comparable to that of 2% trehalose solution.

The effects of trehalose on cognitive function were not associated with a nonspecific effect on general locomotion or exploratory activity, since the open field test did not reveal significant differences between experimental groups in the distance traveled (indicator of total horizontal locomotor activity; *F*(1, 31) < 1) or the number of rearings (indicator of vertical locomotor activity and exploratory activity; *F*(1, 31) = 1.85, *p* > 0.05) (Figure 6A,B). It is worth noting that the parameter of anxiety (time spent in the center of arena) was significantly influenced by trehalose treatment [*F*(1, 30) = 2.9, *p* < 0.05; Figure 6C]. An anxiolytic effect, measured by the prolonged time spent in the center, was observed in the group treated with the highest dosage of trehalose used (4% solution daily).

## 4. Discussion

The present work analyzes several molecular, cellular, and behavioral markers of the processes of neuronal damage and repair during trehalose treatment in an AD model, induced by i.c.v. administration of the Aβ25–35 fragment in mice.

According to the levels of LC3-II in the brain, autophagy is not suppressed in the Aβ25–35-induced AD model, which is consistent with our previous results [12,27] but not with the general concept of impaired autophagy in AD [32]. It should be noted that the central administration of Aβ25–35 alone may result in the significant enhancement of autophagy in the brain [33]. As LC3-II marks autophagosome membranes, this phenomenon can be regarded as a normal early response of autophagy activation to the accumulation of pathological proteins. Here, there was a significant increase in the LC3-II expression in the dentate gyrus and a tendency for an increase within the CA1 and CA3 hippocampal areas and frontal cortex. The smoothing out of the differences between the controls and Aβ-treated group might be explained by the considerable delay between the Aβ administration and the sampling of tissues. Nonetheless, the high initial autophagy level did not prevent the autophagy activation by trehalose treatment. That is also consistent with our previous findings on autophagy activation by rapamycin and trehalose in the brains of mice in Aβ25–35-induced AD models [12]. The therapeutic effect of trehalose is mainly associated with the activation of autophagy and its neuroprotective effect [5]. The manner in which autophagy is regulated by trehalose appears to depend on the activity of the glucose transporter GLUT8, the inhibition of which in the liver leads to the activation of the regulatory kinase AMPK, with the subsequent activation of ULK1 and autophagy [34]. In the brain, trehalose did not inhibit GLUT8 expression and did not activate AMPK phosphorylation but stimulated autophagy, according to a Western blotting of LC3-II [35]. On the other hand, trehalose clearly enhanced AMPK phosphorylation, autophagy, and neuronal viability in a mouse model of Parkinson’s disease [36].

Here, we revealed a dose-dependent enhancement of autophagy in the brain structures with the highest levels at a treatment regimen of a daily intake of 4% trehalose solution. The result obtained appears to differ from some previous findings. In particular, treatment with 2% and 4% trehalose solution similarly enhanced autophagy in the brain tissue in a mouse model of Mn-induced α-synuclein oligomerization [23]. In rats, treatment with 2% and 5% trehalose solution had similar effects on autophagy induction in the brain, while 0.5% solution was ineffective in a AAV1/2-based rat model of Parkinson’s disease [18]. It should be noted that both studies assessed autophagy in the striatum and were performed in Parkinson’s disease-related models. One may suggest structure- and model-dependent effects. An increase in autophagy at high concentrations of trehalose may be related to the augmented penetration of trehalose into the blood due to the saturation of the gut enzyme trehalase, which degrades trehalose [22]. Intestinal trehalase is thought to fail to degrade the abundance of substrate, and hence more trehalose enters the bloodstream, reaches the brain, and causes increased brain responses. In any case, the effect of reducing the increased autophagy due to self-inhibition by its excessive stimulation was not found in the present study. Accordingly, the results on the stimulation of autophagy by 4% trehalose solution in the intermittent mode did not show any advantage over the continuous stimulation by 4% trehalose solution.

The pathogenesis of AD is considered to be largely related to the amyloid cascade, in particular with the accumulation of Aβ and its various aggregated forms [37]. Aβ load could be the result of its increased production or its decreased clearance by the protein quality control systems and, first of all, by autophagy. Earlier, we revealed an intracellular Aβ immunostaining in the Aβ25–35-induced mouse AD model, which could be considered as the initial stages of amyloidogenesis, Aβ seeding and aggregation [12,27]. Here, we also found the augmented levels of intracellular Aβ in mice of the Aβ group. Moreover, the levels of autophagy activity assessed by LC3-II expression were not reduced in these animals. Hence, the Aβ accumulation in this model is associated with the activation of its production, rather than inhibition of its degradation through the autophagy mechanisms in particular. This result is consistent with general patterns of Aβ accumulation in the experimental AD models [38].

Soluble Aβ oligomers are considered as the main neurotoxic forms contributing to the development of AD pathology [39]. When administered to experimental animals, Aβ oligomers produce acute neurotoxicity, including synaptic dysfunctions, and trigger amyloid cascade [40], while the pronounced behavioral and neuromorphological alterations appear in 2–4 weeks after Aβ oligomers injection [41,42]. Here, trehalose dramatically reduced the Aβ levels at all treatment regimens applied. As mice had been treated with trehalose for two weeks starting from the 2 days after Aβ administration, we may conclude that trehalose hampers Aβ accumulation and plaque formation. This agrees with previous reports about the reduction of Aβ accumulation by trehalose both in cellular [9] and in vivo [43,44] AD models. At the same time, we cannot exclude that trehalose treatment might be effective for reverting the already formed amyloid plaques as well, since 1% of trehalose in the drinking water for 3 weeks decreased the number of amyloid plaques in the old (14-month-old) transgenic mice with tauopathy [13]. Although the effects of the trehalose treatment regimens on the autophagy activation differed significantly with the maximal effect of daily intake of 4% trehalose solution, Aβ accumulation was effectively resolved by all the types of treatment. This result might be explained by the involvement of other mechanisms in the decrease in Aβ levels. Trehalose was found to produce neuroprotective effects via different molecular pathways, including its chaperone-like activity, which can prevent the production of different forms of Aβ [5]. Thus, all the regimens studied are sufficient for the treatment of Aβ pathology in the frontal cortex and hippocampus. The strong attenuation of Aβ accumulation may have a curative effect on the AD-like pathology.

The accumulation of Aβ oligomers in the brain provokes the development of neuroinflammation and the activation of microglia, which is a critical step in AD progression and in the amyloid cascade [40]. As in our previous studies [27], the expression of the microglial marker IBA1 was significantly increased in the frontal cortex and hippocampus in mice of the Aβ group that corresponds to the course of acute neuroinflammation. Trehalose treatment in all regimens profoundly attenuated the neuroinflammatory response. Nevertheless, no significant differences were found between the trehalose-treated groups. Taking into account a significant positive correlation between the Aβ and IBA1 levels in the brain structures revealed earlier [27], we suggest that the effect of trehalose on microglia activation is associated with the inhibition of Aβ accumulation, rather than its direct influence.

We also assessed a system immune response by the indices of chronic neuroinflammation known for experimental animals [30] or the blood biomarkers accompanying AD in humans [31]. Here, we studied the levels of inflammatory markers LE and α1-PI. AD patients are characterized by signs of generalized inflammation associated with the activation of acute phase proteins (especially α1-PI) and the decreased (exhausted) functional activity of neutrophils (assessed by LE activity), which are exacerbated by disease progression [31]. In mice with Aβ-induced AD-like pathology, no significant alterations in the levels of LE or α1-PI compared to controls were found. It should be noted that these indices refer to a long-lasting pathological process. It is highly likely that these signs of generalized chronic system inflammation are not reproduced in the Aβ25–35-induced model because of the short-term period of the experiment related to acute neuroinflammation. Trehalose affected the LE activity but not α1-PI levels. However, the effects of trehalose on the activity of LE were inconsistent: consumption of the 2% trehalose solution enhanced the parameter, while intermittent application of the 4% trehalose solution every other day markedly reduced the activity of the enzyme. A decrease in the degranulation activity of peripheral neutrophils (LE) when using trehalose in high doses indicates the possibility of its anti-inflammatory effect. Thus, trehalose could modulate the activity of peripheral immune cells (neutrophils), but the mechanisms of the effect and its relevance to AD course should be further studied in other models with peripheral signs of chronic inflammation.

The neuroprotective potential of trehalose was also evaluated by the restoration of neuronal density in the frontal cortex and hippocampus, as assessed by Nissl staining. The parameter was significantly reduced in the frontal cortex and hippocampal CA1 area in the Aβ group compared to controls, while the trehalose treatment, in all regimens applied, caused its recovery up to the levels of the control group. The results on the restoration of hippocampal neurons by trehalose are in good agreement with previous findings in AD models. For example, trehalose was shown to inhibit the apoptosis of hippocampal neurons in the transgenic APP(swe) murine model of AD [44]. We revealed that all the regimens studied are sufficient for the treatment of neuronal deficits in the frontal cortex and hippocampus at early stages of AD-like pathology. As neuronal dysfunction at the early phases of AD is strongly associated with Aβ pathology [40], the result agrees well with the similar effects of different trehalose regimens on Aβ levels in this model.

The most intriguing was the effect of trehalose treatment on cognitive performance. Long-term fear-associated memory and learning were considerably disturbed in Aβ-treated mice according to the passive avoidance test, which is in good agreement with our previous findings [12,27]. All regimens of trehalose application had a positive effect on cognitive recovery. Moreover, a dose-dependent effect of trehalose was revealed, with maximal values in the group given 4% trehalose solution daily. It is worth noting that the values of step-through latency were only completely restored up to the values of the controls in the “Aβ + 4% T” group. At the same time, experimental groups did not differ significantly in the general locomotion or exploratory activity, as measured in the open field test or by the parameters of mouse activity in the passive avoidance test on habituation and training day. Thus, the effects of trehalose on the cognitive function measured in the passive avoidance test were specific and did not depend on general locomotor or exploratory activity, or the general arousal of animals.

In addition, we revealed an anxiolytic effect of the daily treatment with 4% trehalose solution, assessed by the increased time spent in the center of the open field. Elevated anxiety is tightly associated with AD [45]. However, AD modeling by Aβ administration does not always produce a clear anxiety-like phenotype. It may depend on many factors, including the extent of the disturbances produced by the acute neurotoxic influence of Aβ fragments and possible spontaneous restoration. Here, we did not find anxiogenic behavior in Aβ-treated mice, while in our previous experiments we observed increased anxiety measured in the plus-maze test in Aβ25–35-treated mice, which was attenuated by the combined administration of rapamycin and trehalose but not by the 2% trehalose treatment alone [12]. It is worth noting that 1% of trehalose in the drinking water for 3 weeks reverts anxiety-like phenotypes in the old (14-month-old) transgenic mice with tauopathy [13]. Thus, we suppose that trehalose may produce an anxiolytic effect in AD, but this suggestion should be further confirmed using other AD models with pronounced anxiety phenotype. The results are generally consistent with the notion that trehalose administration attenuates anxiety symptoms and improves memory impairment in mouse models of tau pathology and AD [4].

The high therapeutic efficacy of 4% trehalose intake needs to be further and more broadly explored in the context of the inevitable translational transition to clinical trials. First, the high dosage of trehalose in mice (4% solution daily) should be recalculated for humans. It corresponds to an extremely high single dosage for humans (over 200 g), while doses of approximately 100 g per person per day have been found to be safe [15,46]. Higher doses have not been tested in humans yet. Nevertheless, the tolerability of trehalose by oral intake is high, with rare adverse effects mainly due to the infrequent presence of the bacterium Clostridium difficile in the intestinal microbiota, which proliferates rapidly in the presence of trehalose and causes diarrhea [47]. Concerning the bioavailability and pharmacokinetics of orally administered trehalose, approximately 1% of the consumed trehalose is absorbed into the blood [48,49] (some reports assume higher levels of absorption up to 20% [50]). No more than 1% of the trehalose which enters into the blood from the intestines gets into the brain. When trehalose is administered orally, it is absorbed quite quickly with a maximal concentration in the blood in 30 min, and its removal from the blood ends in 5 h. The bulk of trehalose is digested in the intestine by the enzyme trehalase into two glucose residues. Nevertheless, blood glucose levels rise by only 20% [51] and then decrease below even normal levels [52]. Trehalose does not undergo oxidative metabolism with liver cytochromes.

At the same time, other translational approaches are being developed related to the parenteral administration of the drug [53] and, in particular, intravenous injections of trehalose [54]. The latter allows us to achieve a high concentration of trehalose in the blood at lower dosages applied. On the other hand, the oral administration of trehalose has certain advantages, including the noninvasive method of application and the large accumulated knowledge of the treatment in animals [4]. In any case, our data on the stronger effects of high doses of trehalose on cognitive function and anxiety would be useful for translation of the experimental trehalose treatment to therapy for AD in clinics.

## 5. Conclusions

In summary, we have attempted to evaluate the therapeutic efficacy of trehalose treatment regimens in a mouse model of early stages of Aβ-induced AD-like pathology. We revealed the dose-dependent effects on autophagy activation in the frontal cortex and hippocampus, as well as on the restoration of behavioral disturbances related to cognitive function. A daily intake of 4% trehalose solution for two weeks caused the greatest activation of autophagy, and the complete recovery of the step-through latency on the test day in the passive avoidance test that corresponds to associative long-term memory and learning, while other regimens caused the partial repair of the cognitive function. The restoration of the behavioral deficits related to cognitive impairment is one of the most critical issues while testing drugs in animal AD models. Thus, high doses of trehalose were found to have increased efficacy towards cognitive impairment in a model of early AD-like pathology, and the results indicate the superiority of the 4% daily dosing regimen.

The more pronounced effects of the high trehalose dosage might be attributed to the higher blood levels of the drug because of the intestinal trehalase saturation. Intestinal trehalase is thought to fail to degrade the abundance of substrate, and hence more trehalose enters the bloodstream, reaches the brain, and causes increased brain responses. At a 4% trehalose solution intake in the intermittent mode, this advantageable mechanism seems to be lost, and the effects were similar to daily intake of 2% trehalose solution. According to our calculations, the daily intake dose for 2% trehalose solution in our experiments was approximately 4 g/kg/day, and for 4% trehalose solution it was 8 g/kg/day.

At the same time, the effects of all the treatment regimens studied were similar on the Aβ load, neuroinflammatory response, and neuronal density in the frontal cortex and hippocampus. The trehalose treatment successfully restored these parameters to the levels of the control group. We suggest that those neurodegenerative features are less affected by the pathological processes at early stages of Aβ-induced AD-like pathology, and thus they could be effectively reversed by relatively low doses of trehalose. Moreover, besides autophagy, there are several other important targets and mechanisms related to the neuroprotective activity of trehalose, including its chaperone-like activity, antioxidant activity, and its modulation of ubiquitin-proteasome system [5]. This multi-target neuroprotective activity of trehalose appears to be highly effective for the treatment of Aβ pathology and related disturbances. These findings could be taken into account for translational studies and the development of clinical approaches for AD therapy using trehalose.

## Figures and Tables

**Figure 1 pharmaceutics-16-00813-f001:**
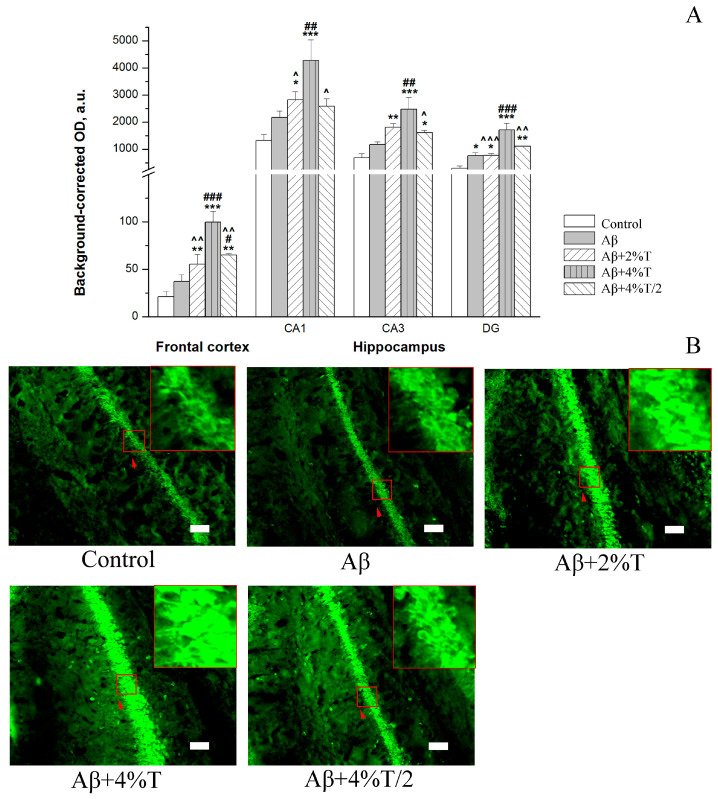
The effect of treatment with trehalose (2% or 4% solution daily (“Aβ + 2% T” or “Aβ + 4% T”) or 4% solution every other day (“Aβ + 4% T/2”), 14 days) on autophagy activity, quantified as immunofluorescence of an autophagy marker (LC3-II) in the brain of mice with Aβ-induced AD-like pathology. To set up the AD model, mice were given bilateral i.c.v. injections of Aβ25–35. The control group received vehicle (sterile water) i.c.v. injections. The treatment with trehalose was started 2 days after the i.c.v. Aβ25–35 administration. (**A**) Quantitative results. The data are expressed as the mean ± SEM of the values obtained in an independent group of animals (n = 3 per group). Statistically significant differences: * *p* < 0.05, ** *p* < 0.01, *** *p* < 0.001 vs. control group; # *p* < 0.05, ## *p* < 0.01, ### *p* < 0.001 vs. Aβ group; ^ *p* < 0.05, ^^ *p* < 0.01, ^^^ *p* < 0.001 vs. “Aβ + 4% T” group. (**B**) LC3-II immunofluorescence in the CA1 hippocampal area. The fluorescence images were finally obtained by an Axioplan 2 microscope. Magnification: ×200; scale bar (white rectangle), 50 μm. High zoom images of areas indicated by red arrows within red squares with LC3-II-positive cells are shown in the insets.

**Figure 2 pharmaceutics-16-00813-f002:**
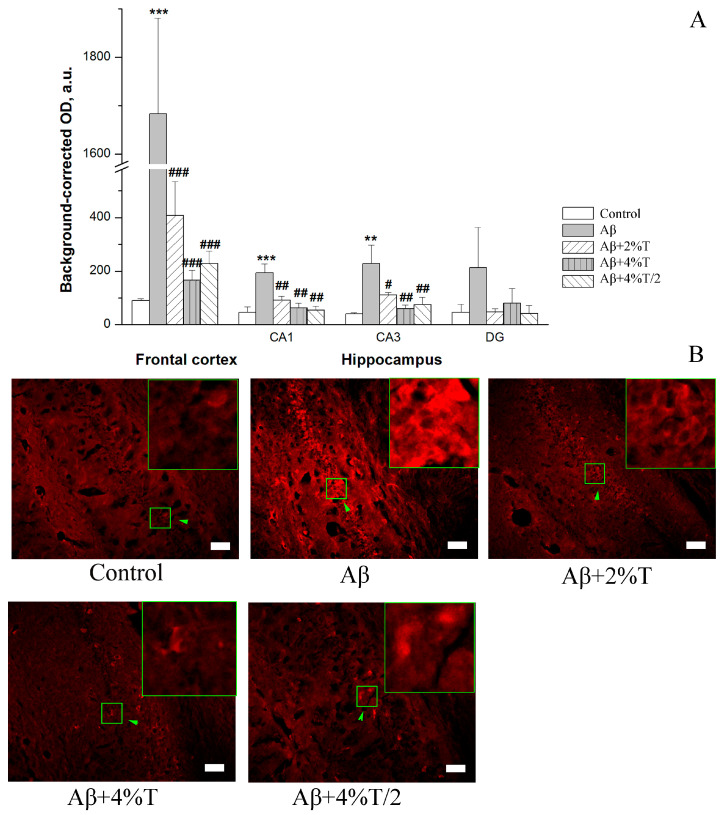
The effect of treatment with trehalose (2% or 4% solution daily (“Aβ + 2% T” or “Aβ + 4% T”) or 4% solution every other day (“Aβ + 4% T/2”), 14 days) on Aβ accumulation, quantified as immunofluorescence of Aβ in the brain of mice with Aβ-induced AD-like pathology. To set up the AD model, mice were given bilateral i.c.v. injections of Aβ25–35. The control group received vehicle (sterile water) i.c.v. injections. The treatment with trehalose was started 2 days after the i.c.v. Aβ25–35 administration. (**A**) Quantitative results. The data are expressed as the mean ± SEM of the values obtained in an independent group of animals (n = 3 to 4 per group). Statistically significant differences: ** *p* < 0.01, *** *p* < 0.001 vs. control group; # *p* < 0.05, ## *p* < 0.01, ### *p* < 0.001 vs. Aβ group. (**B**) Aβ immunofluorescence in the CA1 hippocampal area. The fluorescence images were finally obtained by an Axioplan 2 microscope. Magnification: ×200; scale bar (white rectangle), 50 μm. High zoom images of areas indicated by green arrows within green squares with intracellular Aβ load are shown in the insets.

**Figure 3 pharmaceutics-16-00813-f003:**
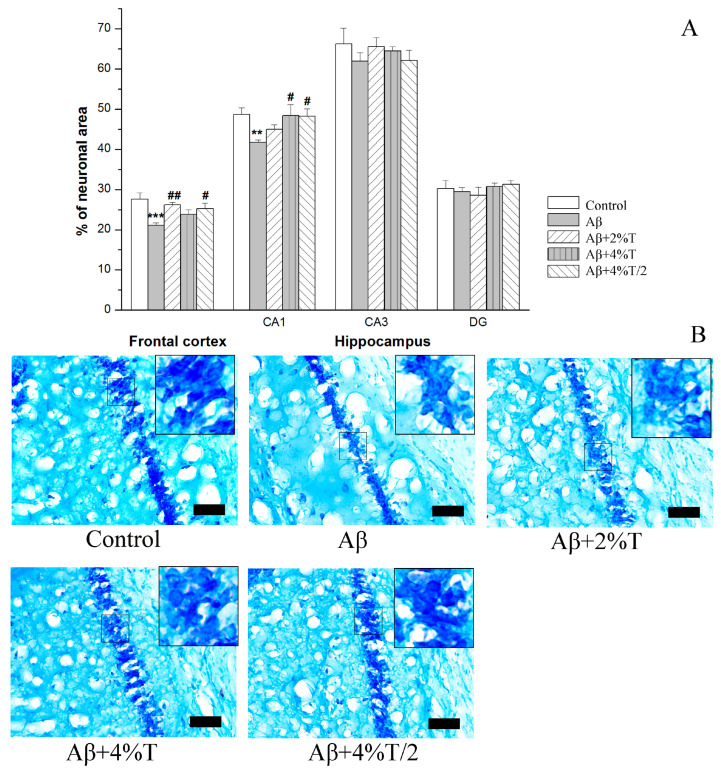
The effect of treatment with trehalose (2% or 4% solution daily (“Aβ + 2% T” or “Aβ + 4% T”) or 4% solution every other day (“Aβ + 4% T/2”), 14 days) on neuronal density, measured as the area occupied by Nissl-stained cells in the brain of mice with Aβ-induced AD-like pathology. To set up the AD model, mice were given bilateral i.c.v. injections of Aβ25–35. The control group received vehicle (sterile water) i.c.v. injections. The treatment with trehalose was started 2 days after the i.c.v. Aβ25–35 administration. (**A**) Quantitative results. The data are expressed as the mean ± SEM of the values obtained in an independent group of animals (n = 3 to 5 per group). Statistically significant differences: ** *p* < 0.01, *** *p* < 0.001 vs. control group; # *p* < 0.05, ## *p* < 0.01 vs. Aβ group. (**B**) Nissl-stained pyramidal neurons in the CA1 hippocampal area; scale bar (black rectangle), 50 μm. High zoom images of areas within small squares with Nissl-stained pyramidal neurons are shown in the insets.

**Figure 4 pharmaceutics-16-00813-f004:**
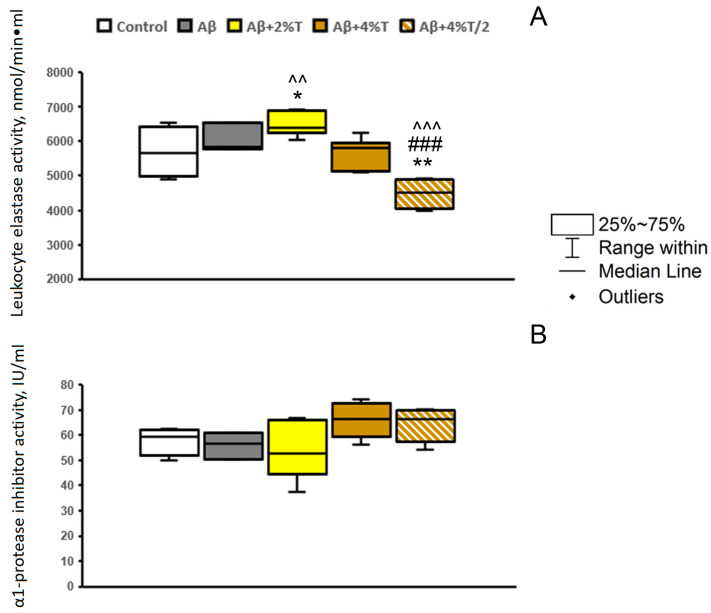
The effect of treatment with trehalose (2% or 4% solution daily (“Aβ + 2% T” or “Aβ + 4% T”) or 4% solution every other day (“Aβ + 4% T/2”), 14 days) on LE activity (**A**) and α1-PI activity (**B**) in the serum in mice with Aβ-induced AD-like pathology. To set up the AD model, mice were given bilateral i.c.v. injections of Aβ25–35. The control group received vehicle (sterile water) i.c.v. injections. The treatment with trehalose was started 2 days after the i.c.v. Aβ25–35 administration. The data are expressed as the median and interquartile range (Q1; Q3) of the values obtained in an independent group of animals (n = 3 to 6 per group). Statistically significant differences: * *p* < 0.05, ** *p* < 0.01 vs. control group; ### *p* < 0.001 vs. Aβ group; ^^ *p* < 0.01, ^^^ *p* < 0.001 vs. “Aβ + 4% T” group.

**Figure 5 pharmaceutics-16-00813-f005:**
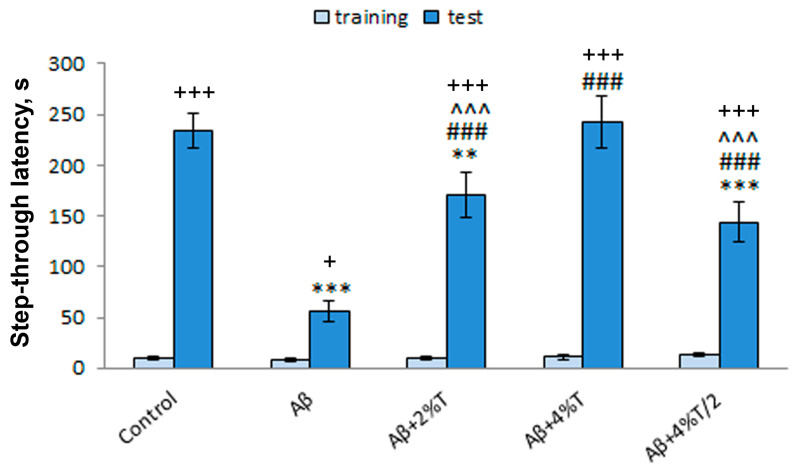
The effect of treatment with trehalose (2% or 4% solution daily (“Aβ + 2% T” or “Aβ + 4% T”) or 4% solution every other day (“Aβ + 4% T/2”), 14 days) on cognitive function (passive avoidance learning) in mice with Aβ-induced AD-like pathology. To set up the AD model, mice were given bilateral i.c.v. injections of Aβ25–35. The control group received vehicle (sterile water) i.c.v. injections. The treatment with trehalose was started 2 days after the i.c.v. Aβ25–35 administration. The data are expressed as the mean ± SEM of the values obtained in an independent group of animals (n = 6 per group). Statistically significant differences: + *p* < 0.05, +++ *p* < 0.001 vs. the values of the same group on training day; ** *p* < 0.01, *** *p* < 0.001 vs. control group; ### *p* < 0.001 vs. Aβ group; ^^^ *p* < 0.001 vs. “Aβ + 4% T” group.

**Figure 6 pharmaceutics-16-00813-f006:**
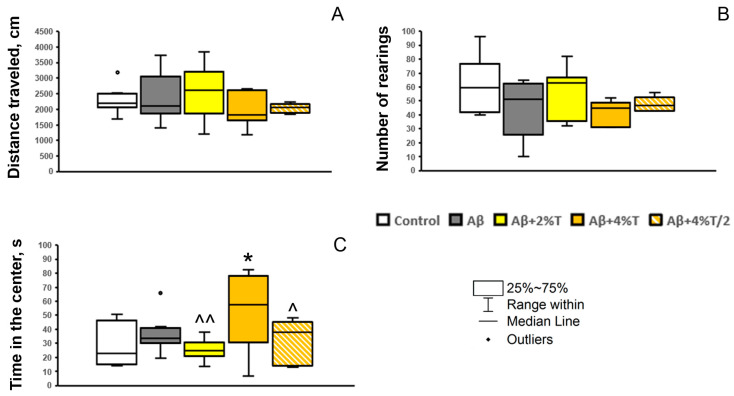
The effect of treatment with trehalose (2% or 4% solution daily (“Aβ + 2% T” or “Aβ + 4% T”) or 4% solution every other day (“Aβ + 4% T/2”), 14 days) on the behavior in the open field test in mice with Aβ-induced AD-like pathology. To set up the AD model, mice were given bilateral i.c.v. injections of Aβ25–35. The control group received vehicle (sterile water) i.c.v. injections. The treatment with trehalose was started 2 days after the i.c.v. Aβ25–35 administration. (**A**) distance traveled (cm), an index of locomotor activity; (**B**) a number of rearing bouts, an index of vertical locomotor activity and exploratory activity; (**C**) time spent in the center (s), an index of anxiety. The data are expressed as the median and interquartile range (Q1; Q3) of the values obtained in an independent group of animals (n = 6 to 9 per group). Statistically significant differences: * *p* < 0.05 vs. control group; ^ *p* < 0.05, ^^ *p* < 0.01 vs. “Aβ + 4% T” group.

## Data Availability

The raw data supporting the conclusions of this article will be made available by the authors on request.

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
