# Peer review of "Alimentary Treatment with Trehalose in a Pharmacological Model of Alzheimer’s Disease in Mice: Effects of Different Dosages and Treatment Regimens"

_pharmaceutics, 2024, doi:10.3390/pharmaceutics16060813_

Round 1
Reviewer 1 Report
Comments and Suggestions for Authors
This article describes the use of various concentrations of trehalose in solution to treat Alzheimers disease symptoms in mice. The article is well structured and the results are presented and discussed clearly. Below are a few questions and comments regarding the results:
- Can you comment on why the LE activity was affected so inconsistently?
- Although you see increased expression of the autophagy marker, reduced Ab accumulation and an improvement in learning (step-through latency) you do not observe significant influence of the trehalose treatment on the inflammatory markers (LE and a1-PI) or other behavioral tests. Can you comment on this? Why are the improvements on a cellular level not translated to behavioral changes?
- you mention an anxiolytic effect of the 4% trehalose solution - however you do not see an increase in anxiety in the Ab-induced group compared to control animal - so it is hard to conclude that trehalose attenuates anxiety symptoms in this AD model
- You mention in the discussion that according to the LC3-II levels in the brain you do not seem to have reduced autophagy in your Ab25-35-induced model compared to literature. Can you comment on why you think this is? How this could be affecting your results?
- It would be very interesting to see the effects of this treatment on a more advanced disease model (longer progression).
- It would be very interesting to see how long the effects of this treatment last - do they need to be continued for disease remission? Will the disease symptoms return if the treatment is interrupted?
Author Response
We would like to thank the Reviewer for his/her valuable comments and suggestions. We greatly appreciate the high esteem of our study. We have thoroughly revised the manuscript considering all the comments, which helped us to improve the manuscript. All major corrections made in the text are highlighted with green color. We believe that the revised version would be more clear and interesting for the readership of the journal.
- Can you comment on why the LE activity was affected so inconsistently?
We suppose that neuroinflammation in the experimental mice did not last long enough to affect the peripheral immune processes and markers. However, a decrease in the degranulation activity of peripheral neutrophils (LE) when using trehalose in high doses indicates the possibility of its anti-inflammatory effect (P. 12). In cases of chronic neuroinflammation, this effect is expected to be more pronounced.
- Although you see increased expression of the autophagy marker, reduced Ab accumulation and an improvement in learning (step-through latency) you do not observe significant influence of the trehalose treatment on the inflammatory markers (LE and a1-PI) or other behavioral tests. Can you comment on this? Why are the improvements on a cellular level not translated to behavioral changes?
Regarding LE and PI, modeling of AD with a single administration of an amyloid fragment does not appear to produce a chronic system inflammation which is observed in AD and could be affected by trehalose treatment. Nevertheless, a decrease in the degranulation activity of peripheral neutrophils (LE) when using trehalose in high doses indicates the possibility of its anti-inflammatory effect. This agrees with the inhibitory effect of trehalose on microglia activation in the brain.
Concerning behavioral changes, there was a clear dose-dependent effect of trehalose on the learning and memory in the passive avoidance test. These parameters were significantly disturbed in the Ab-induced model and restored by the trehalose treatment. So, there is a good agreement between cellular and behavioral levels. On the other hand, locomotor activity measured in the open field test was not profoundly affected by the Ab-induced model nor by trehalose treatment. Hence, the effects of trehalose seem to be specific and associated with the brain areas related to cognitive function (the frontal cortex and hippocampus).
- you mention an anxiolytic effect of the 4% trehalose solution - however you do not see an increase in anxiety in the Ab-induced group compared to control animal - so it is hard to conclude that trehalose attenuates anxiety symptoms in this AD model
Elevated anxiety is tightly associated with AD (Pietrzak et al., 2015). However, AD modeling by Ab administration does not always produce clear anxiety-like phenotype. It may depend on many factors including the extent of disturbances produced by the acute neurotoxic influence of Ab fragments and possible spontaneous restoration. In our previous experiments, we observed increased anxiety measured in the plus-maze test in mice which was attenuated by the combined administration of rapamycin and trehalose but not by the 2% trehalose treatment alone (Pupyshev et al., 2022). Noteworthy, 1% of trehalose in the drinking water for 3 weeks reverts anxiety-like phenotypes in the old (14-month-old) transgenic mice with tauopathy (Rodríguez-Navarro et al., 2010). Thus, we suppose that trehalose has anxiolytic properties in AD but this suggestion should be further confirmed using other AD models with pronounced anxiety phenotype. We revised discussion of this part (P. 13).
- You mention in the discussion that according to the LC3-II levels in the brain you do not seem to have reduced autophagy in your Ab25-35-induced model compared to literature. Can you comment on why you think this is? How this could be affecting your results?
Central administration of Ab25–35 alone may result in significant enhancement of autophagy in the brain (Fan et al., 2015; Korolenko et al., 2019). This phenomenon may be regarded as a normal early response to the accumulation of pathological proteins. Here, there was a significant increase in the LC3-II expression in the dentate gyrus and a tendency for an increase within the CA1 and CA3 hippocampal areas and frontal cortex that is consistent with our previous findings (Pupyshev et al., 2022; Belichenko et al., 2023). The smoothing out of the differences between the controls and Aβ-treated group might be explained by the considerable delay between the Aβ administration and the sampling of tissues. Nonetheless, the high initial autophagy level did not prevent from the autophagy activation by trehalose treatment. That is also consistent with our previous findings on autophagy activation by rapamycin and trehalose in the brain of mice in Aβ25–35-induced AD model (Pupyshev et al., 2022). (P. 11)
- It would be very interesting to see the effects of this treatment on a more advanced disease model (longer progression).
We completely agree with this point. We have already performed an experiment on a transgenic mouse model of AD using age-dependent treatment of mice with trehalose. The results are being processed and will be published in the next paper soon.
- It would be very interesting to see how long the effects of this treatment last - do they need to be continued for disease remission? Will the disease symptoms return if the treatment is interrupted?
The question is certainly interesting and needs special study. We may state that trehalose effects last for at least 1-2 weeks as behavioral testing and brain sampling were performed within this period after the drug withdrawal in the present study or in our previous experiments (Pupyshev et al., 2022; Belichenko et al., 2023).
Reviewer 2 Report
Comments and Suggestions for Authors
The current manuscript is significant as it investigates in vivo dosing regimens for trehalose as a promising agent in the treatment of neurodegenerative diseases. The manuscript is well-written, and the Authors have provided experimental details. However, there are some concerns which require the Authors’ attention as follows,
1. In section 2.1, how the animal trehalose consumption was monitored? Did the Authors measured the volume of water consumed by the animals?
2. For the enzyme assays mentioned in sections 2.4 and 2.5, it is not clear whether positive and negative controls were used for these experiments and no reference to sources which describe the assay in detail (e.g. assay calibration and validation).
3. For figure 1, the small squares around the zoomed in areas are hard to see, would it be possible to make it more clear or add arrow to clarify its position? Also, would it be possible to indicate how much magnification was done?
5. Based on the study results, trehalose regimen of 4% daily for two weeks achieved higher LC3-II (autophagy marker) and marginally higher behavioural change, while all other markers were not affected by regimen except for LE activity which was markedly reduced by intermittent application of 4% trehalose solution every other day. I am not sure that these results indicate the superiority of the 4% daily dosing regiment. Would the Authors be able to further discuss this point in the discussion section?
6. In the conclusion section (section 5), in lines 486 to 488, the Authors are comparing their results to other published results. I do not think that this fits in the conclusion section as it is better to move this part to the discussion section in order to keep the conclusion section short and include only the main points which we can conclude from the study.
Comments on the Quality of English LanguageIn page 8, line 264, the statement “Thus, inhibitory influence of trehalose on the brain neuroinflammation was found” is not clear, would the Authors be able to clarify this statement.
Author Response
We would like to thank the Reviewer for his/her valuable comments and suggestions. We greatly appreciate the high esteem of our study. We have thoroughly revised the manuscript considering all the comments, which helped us to improve the manuscript. All major corrections made in the text are highlighted with green color. We believe that the revised version would be more clear and interesting for the readership of the journal.
- In section 2.1, how the animal trehalose consumption was monitored? Did the Authors measured the volume of water consumed by the animals?
The practice of treating mice with trehalose using its 1-5% solutions as drinking is widespread. Mice of such body mass (25 g approximately) drink about 5-6 ml of trehalose solution per day. In our experiments, the amount of liquid drunk was determined. Male C57BL/6 mice consumed 5.63 + 0.18 ml of water (mouse body mass was 25.74 + 0.31 g) or 5.14 + 0.30 ml of trehalose solution (p>0.05; mouse body mass was 25.04+0.32 g). (P.3)
- For the enzyme assays mentioned in sections 2.4 and 2.5, it is not clear whether positive and negative controls were used for these experiments and no reference to sources which describe the assay in detail (e.g. assay calibration and validation).
We corrected the description of the assays and add a reference to the detailed description of the method (P. 4).
- For figure 1, the small squares around the zoomed in areas are hard to see, would it be possible to make it more clear or add arrow to clarify its position? Also, would it be possible to indicate how much magnification was done?
Figures 1 and 2 were revised according to the comment.
- Based on the study results, trehalose regimen of 4% daily for two weeks achieved higher LC3-II (autophagy marker) and marginally higher behavioural change, while all other markers were not affected by regimen except for LE activity which was markedly reduced by intermittent application of 4% trehalose solution every other day. I am not sure that these results indicate the superiority of the 4% daily dosing regiment. Would the Authors be able to further discuss this point in the discussion section?
We suppose that restoration of behavioral deficits related to cognitive impairment is one of the most critical issues while testing drugs in animal AD models. 4% daily dosing regimen completely restored associative memory disturbances in AD mice while other regimens caused partial repair of the function. (P.14, Conclusions)
- In the conclusion section (section 5), in lines 486 to 488, the Authors are comparing their results to other published results. I do not think that this fits in the conclusion section as it is better to move this part to the discussion section in order to keep the conclusion section short and include only the main points which we can conclude from the study.
We removed this part from the Discussion and revised the text.
Comments on the Quality of English Language
In page 8, line 264, the statement “Thus, inhibitory influence of trehalose on the brain neuroinflammation was found” is not clear, would the Authors be able to clarify this statement.
We corrected the statement as following:
Thus, an inhibitory effect of trehalose on neuroinflammation in the brain was found.
Reviewer 3 Report
Comments and Suggestions for Authors
The manuscript by Pupyshev and coworkers discusses the use of dietary threalose treatment with three different drink solution administration regimens (2% daily, 4% daily or 4% intermittent) in muriune model of Alzheimer's Disease. The outcome is evaluated based on behavioural testing as well as neuromorphological analyses and in vitro evaluation of neuroinflammation markers.
The manuscript is clear, relevant for the field and presented in a well-structured manner. The manuscript is scientifically sound and the experimental design is appropriate to test the hypothesis. Conclusions are consistent with the evidence and arguments presented. Cited references are mostly recent and relevant publications.
However, the authors should describe if, since trehalose treatment was started 2 days after Aβ25-35 injection, this time was sufficient to induce the pathology. Moreover, in the discussion section the authors suggest that trehalose reduced accumulation of Aβ . Is there any evidence that trehalose could reverse the process of amyloid deposits formation? Does trehalose serve only as preventive treatment to avoid future plaque formation or can it be envisaged as treatment for reverting the already formed amyloid plaques? Please discuss this scenery.
Moreover, it would be important to consider and briefly discuss the bioavalability of oral administed trehalose, including pharmacokinetics and pharmacodynamics of such molecule.
Finally, the authors should expand acronyms the first time they appear in the text, not only in the section titles (paragraphs 2.4 and 2.5).
Author Response
We would like to thank the Reviewer for his/her valuable comments and suggestions. We greatly appreciate the high esteem of our study. We have thoroughly revised the manuscript considering all the comments, which helped us to improve the manuscript. All major corrections made in the text are highlighted with green color. We believe that the revised version would be more clear and interesting for the readership of the journal.
However, the authors should describe if, since trehalose treatment was started 2 days after Aβ25-35 injection, this time was sufficient to induce the pathology. Moreover, in the discussion section the authors suggest that trehalose reduced accumulation of Aβ . Is there any evidence that trehalose could reverse the process of amyloid deposits formation? Does trehalose serve only as preventive treatment to avoid future plaque formation or can it be envisaged as treatment for reverting the already formed amyloid plaques? Please discuss this scenery.
A dominant hypothesis of AD pathogenesis is the concept of amyloid cascade with a central causative role of amyloidogenic protein Aβ for AD progression. Hence, a central administration of Aβ-containing solutions is widely used to induce AD-like pathology in animal models. Soluble Aβ oligomers (AβO) are considered as the main neurotoxic forms contributing mostly to the development of AD pathology (Marshall et al., 2016). When administered to experimental animals, AβO produce acute neurotoxicity including synaptic dysfunctions and trigger amyloid cascade (Selkoe, Hardy, 2016). Nevertheless, the pronounced behavioral and neuromorphological alterations appear in 2-4 weeks after AβO injection (Park et al., 2011; Choi et al., 2013). As mice had been treated with trehalose for two weeks starting from the 2 days after AβO administration, we may conclude that trehalose hampers Aβ accumulation and plaque formation. This agrees with previous reports that trehalose reduces the accumulation of amyloid (Du et al., 2013; Nien et al., 2016) and its aggregates (Perucho et al., 2012). At the same time, we cannot exclude that trehalose treatment might be effective for reverting the already formed amyloid plaques as well since 1% of trehalose in the drinking water for 3 weeks decreased the number of amyloid plaques in the old (14-month-old) transgenic mice with tauopathy (Rodríguez-Navarro et al., 2010). (P. 11-12)
Moreover, it would be important to consider and briefly discuss the bioavalability of oral administed trehalose, including pharmacokinetics and pharmacodynamics of such molecule.
Concerning bioavalability and pharmacokinetics of orally administered trehalose, mice drink about 6 ml of trehalose solution per day and about 1% of the consumed trehalose is absorbed into the blood (van Elburg et al., 1995; Kamiya et al., 2004) (some reports assume higher levels of absorption up to 20% (Di Rienzi, Britton, 2020)). No more than 1% of the trehalose which enters into the blood from the intestines gets into the brain. When trehalose is administered per os, it is absorbed quite quickly with a maximal concentration in the blood in 30 min, and its removal from the blood ends in 5 h. The bulk of trehalose is digested in the intestine by the enzyme trehalase into two glucose residues. Nevertheless, blood glucose levels rise by only 20% (Yoshizane et al., 2017) and then decreases even below normal levels (Korolenko et al., 2022). Trehalose does not undergo oxidative metabolism with liver cytochromes. (P. 13)
Pharmacodynamics related to the main molecular mechanism of autophagy activation is briefly discussed on P. 11. Other molecular mechanisms and targets of trehalose are mentioned here and discussed in details in our recent review (Pupyshev et al., Pharmacol Res., 2022, doi: 10.1016/j.phrs.2022.106373).
Finally, the authors should expand acronyms the first time they appear in the text, not only in the section titles (paragraphs 2.4 and 2.5).
OK, corrected.